# Representative Test Material for Validation of Density Separation as Part of Microplastic Quantification in Drinking Water

**DOI:** 10.3390/polym17040526

**Published:** 2025-02-18

**Authors:** Jessica Ponti, J. Francisco Barbosa-de-Bessa, Dora Mehn, Guillaume Bucher, Gabriella F. Schirinzi, Francesco Fumagalli, Douglas Gilliland

**Affiliations:** European Commission, Joint Research Centre (JRC), 21027 Ispra, Italy; jessica.ponti@ec.europa.eu (J.P.); guillaume.bucher@ec.europa.eu (G.B.); gabriella.schirinzi@ec.europa.eu (G.F.S.); francesco-sirio.fumagalli@ec.europa.eu (F.F.); douglas.gilliland@ec.europa.eu (D.G.)

**Keywords:** microplastics, PVC, representative test material, PVC-JRCNM70508a, density separation, drinking water

## Abstract

The evolving regulatory landscape for microplastics—including the European Union’s Drinking Water Directive—underscores the importance of addressing the analytics of emerging contaminants in water, ensuring public health protection, and fostering scientific advancements in environmental monitoring. This work aims to contribute to these advancements by sharing the strategy of test material selection and characterisation for the validation of sample treatment protocols. The article describes a PVC-based representative test material of industrial origin, its physicochemical characterisation, and its application in density separation procedure evaluation, compatibly with the European Commission’s recommendations for quantifying microplastics in water for human consumption. The work shares our protocol for the durable fluorescent labelling of microplastic particles and for the centrifugal density separation of microplastics from other particulate contaminants in drinking water samples. It reports density and viscosity values for the zinc chloride solutions used to feed the theoretical calculations and recovery values achieved with the presented density separation protocol.

## 1. Introduction

Microplastics (MPs), tiny plastic particles less than 5 millimetres in size, have become a growing environmental and public health concern due to their widespread presence, including in drinking water [1,2]. The potential risks associated with microplastic ingestion have prompted regulatory bodies worldwide to establish guidelines and requirements for their monitoring and control [3]. In Europe, the regulatory framework addressing microplastics in drinking water falls under the European Union’s Drinking Water Directive (EU DW Directive). The Directive, which was most recently updated in 2020 (Directive (EU) 2020/2184) and entered into force on the 12 January 2021, aims to ensure that water intended for human consumption is safe and clean, setting standards for various contaminants, including emerging pollutants like microplastics [4]. Under the EU DW Directive, Member States are required to monitor and assess the quality of drinking water against a set of parameters that include chemical, physical, and microbiological criteria. While MPs were not explicitly listed in the earlier versions of the Directive, the 2020 update reflects an increasing awareness of the need to address such contaminants. The Directive empowered the European Commission to develop a consistent methodology for their detection and quantification in water for human consumption. The methodology proposed by the European Commission, publicly available since March 2024, involves a harmonised sampling procedure and foresees the application of sample preparation steps followed by the application of analytical techniques such as infrared spectromicroscopy (µFT-IR) or Raman spectromicroscopy (µRaman) [5]. These methods are designed to identify microplastics, differentiating them from other particles, and to provide a number-based quantification [6,7].

The actual levels of the MP contamination of drinking water in Europe was found to fall in the 0.1–1000 particle/m^3^ range [8]. This very wide range documented in the literature and the variability of instrumental solutions for spectromicroscopic analysis requires a methodology that is flexible enough to allow the adjustment of sample preparation steps depending on the composition of the matrix (the presence of particles other than microplastics) and the actual identification protocol [9]. However, the freedom that allows laboratories to develop tailored sample preparation protocols is linked to the need for the validation of these protocols. The missing piece in this process is a reliable, representative test material that could prove the validity of the selected sample preparation protocol through recovery test(s). Unfortunately, there is a lack of standards/test materials available on the market that are well characterised regarding their size, shape/morphology (spheres, fibres, beads, or films), and polymer type. In addition, there is a lack of well-characterised MP materials representative of industrial-scale production that could be widely distributed for research purposes. For this reason, the JRC Nanomaterials Repository has undertaken a feasibility study on MPs following the already applied practice where each type of material has been sourced as a large single batch to be subsampled into individual vials available for benchmarking in research and regulatory studies. 

Most of the commercially available MPs considered as “standards” are spherical polystyrene (PS) or—less frequently—polyethylene (PE) or poly-methyl methacrylate (PMMA) beads with a density of about 0.8–1.2 g/mL. These few readily available model particles usually do not fulfil the requirements for standard reference materials [10], do not cover the wide range of the physicochemical properties of plastics developed and used in industrial applications (including polymers with densities up to 2.2 g/mL), and do not well represent the irregularly shaped secondary microplastic particles that may be released from these materials. Reference material development is ongoing at various public institutions, also considering the use of irregular-shaped particles, but these materials often are produced by the cryo-milling of larger plastic items and will be available in lower volumes after the long period of evaluation that is needed to verify their homogeneity and stability [11]. Both shape and density are important parameters that might affect the recovery of particles during the sample preparation process. Spherical particles are supposed to adsorb less strongly on filter and vial surfaces than their irregular-shaped counterparts [12]. Density will determine if the particles sediment or float in the various aqueous solutions applied to remove contaminating particles and if they will accumulate closer to the vial bottom or to the liquid/air interface. Particle density becomes particularly important if a density separation step is included in the sample preparation process. Density separation might be a non-trivial element of sample preparation in the case of drinking water samples [13]. Indeed, in the existing literature, it is applied typically when particles are collected from larger volumes of water or when the water contains a high concentration of “other” particles that are not chemically separable from plastics [14]. Silica is such an example, which might appear not only in the form of irregular-shaped sand particles, but also in the form of needle- or fibre-like structures with a biological origin (diatoms) [15]. 

Having in mind these challenges and considerations, hereafter we present the detailed characterisation of a representative test material proposed for method validation, with a particular focus on its use in the validation of density separation. Polyvinyl chloride (PVC) was the polymer chosen, due to several key characteristics such as its widespread use worldwide (pipes, packaging, etc.) [16,17] and consequently its potential presence as secondary microplastic generated from waste in aquatic and terrestrial environments [18,19,20,21], its distinct physical and chemical properties, such as its resistance to many solvents and chemicals, its unique IR and Raman spectra, and—last but not least—its high density compared to many other plastics. Moreover, the PVC material chosen has a significant feature that is of industrial origin: it is a “real world” material used as a starting material to be mixed with other components, for the manufacturing of profiles produced by the extrusion process. It is also representative test sample in terms of its broad size distribution and irregularly shaped particles. 

Our work presents not only the strategy we applied during the selection and careful size distribution and composition characterisation of the test material, but also the applicability of durable fluorescent labelling for these particles, with liquid medium density and viscosity as the input parameters and calculations considered when developing and validating our centrifugation-based density separation method, as well as the results when applying the density centrifugation separation to a variety of different MP types. 

## 2. Materials and Methods

### 2.1. Materials

A batch of powdered polyvinyl chloride (PVC) for use in extrusion moulding was selected and it received the code PVC-JRCNM70508a at our laboratory (JRC Nanomaterials Repository facility, European Commission, Joint Research Centre (JRC), Ispra, Italy). Fluorescent polystyrene (PS) microspheres (142 and 51 µm) were purchased from Thermo-Fisher Scientific (Monza, Italy) and Nylon fibres (25 × 900 µm) from Flocking Ltd. (Heanor, United Kingdom).

Absolute ethanol was purchased from Supelco; hydrochloric acid fuming 37% and concentrated nitric acid (69% HNO_3_ TraceSELECT) from Fluka Analytical (Darmstadt, Germany); and Triton X-100, Periodic Table Mix 1 (33 elements at 10 mg/L), Periodic Table Mix 2 (17 elements at 10 mg/L), and Gallium Standard (1000 mg/L) for ICP from Sigma-Aldrich (Merck, Darmstadt, Germany). In order to prepare the density separation solutions, zinc chloride (98% ZnCl_2_, Merck, Darmstadt, Germany) was used and for the particle staining process and Nile red (Merck, Darmstadt, Germany) was the chosen dye. All other chemicals were purchased from Sigma Aldrich (Merck, Darmstadt, Germany).

All lab ware used was washed with detergent-containing water and rinsed with MilliQ water. Ultrapure water was produced by a MilliQ IQ 7003 machine (Merck, Darmstadt, Germany) equipped with a Millipak 0.22 µm Merck Filter, and this was also the water source employed for the experiments. Throughout the methodology, the particles were also traced by naked eye observation using an Alonefire UV 365 NM 10W LAMP X90 (Shenzhen Shiwang Lighting Co., Shenzen, China) that helped to identify where/if particle losses were occurring.

### 2.2. Identification by Raman and FT-IR Spectromicroscopy

Fourier transform infrared (FT-IR) spectromicroscopy analysis was performed using a Hyperion 3000 FT-IR INVENIO instrument (Bruker Optics GmbH, Ettlingen, Germany), running 32 scans at 4 cm^−1^ of resolution in the range of 4000–400 cm^−1^. Particles were placed on transparent calcium fluoride windows (Crystran Ltd., Poole, UK) and their spectra measured in transmission mode. The spectra then underwent a match search using an FT-IR database for polymers provided by the instrument supplier and the results were subsequently verified using the single-spectrum search option of siMPle [22].

The µRaman analysis was carried out using an alpha300 confocal Raman microscope (WITec, Ulm, Germany) applying a 532 nm laser. Particles were deposited on a clean, polished silicon surface. Measurements were performed using a grating: T1: 600 g·mm^−1^ BLZ = 500 nm, with a 10 s integration time and by averaging 15 accumulations. The spectral recognition was performed based on comparison with a library of literature origin [23].

### 2.3. Elemental Composition

Experiments to characterise the elemental composition of the materials were carried out using different analytical techniques including Inductively Coupled Plasma Mass spectrometry (ICP-MS), Total Reflection X-ray Fluorescence (TXRF), and Transmission electron microscopy–Energy Dispersive X-ray Spectroscopy (TEM-EDX).

For the TXRF analysis, an S4 T-Star TXRF spectrometer (Bruker Nano GmbH, Berlin, Germany) was used equipped with a Mo X-ray source. About 10 mg of PVC powder was mixed with 1 mL of a solution of gallium internal standard (1 mg/L) and Triton X-100 (0.1%). Then, 5 µL of the mixture was pipetted on an acrylic sample holder (B-A20V11, Bruker Nano GmbH, Berlin, Germany). After careful drying, the samples were transferred to the instrument and the TXRF spectra were collected using a 600 s integration time. Data were analysed considering the possible presence of 18 various elements (Appendix A). 

ICP-MS multi-elemental (39 elements) semi-quantitative analysis was performed with a 7700× ICP-MS (Agilent Technologies, Santa Clara, CA, USA) operated in helium collision mode (detailed instrument configuration available in Appendix A) after microwave-assisted acidic digestion of the PVC powder. Briefly, 20.3 mg of PVC powder was transferred to a 35 mL clean glass digestion vessel and 4 mL of concentrated nitric acid was added. Microwave-assisted digestion was performed using a Discover SP-D microwave digestion system (CEM, Cologno Al-Serio, Italy) with a maximum power of 300 W and the following 5-step program: (i) 5 min ramp to 100 °C, (ii) 10 min hold at 100 °C, (iii) 5 min ramp to 220 °C, (iv) 20 min hold at 220 °C, (v) cooling down to 40 °C (ca. 10 min). After cooling to room temperature, the digest (clear and transparent) was quantitatively transferred to a 50 mL polypropylene centrifuge tube and ultrapure water was added to reach a final volume of 50 mL prior to ICP-MS analysis. Instrument performance (sensitivity; oxide and doubly charged ion ratios) was checked daily after optimisation of the measurement conditions using a standard built-in software procedure and a multi-elemental tuning solution. The PVC digested was analysed against a multi-elemental calibration curve consisting of 3 points (0–20–40 µg/L) in 8% HNO_3_.

### 2.4. Fractionation (Sieving)

The selected PVC sample, PVC-JRCNM70508a, was fractionated to different size ranges, based on selected test sieves (20–40, 40–80, 80–160, 160–315, and 315–600 µm), using the Retsch sieving (Retsch GmbH AS200 Basic 60Hz, Torre Boldone, Italy), and a mass-based analysis of the fraction was carried out using a Mettler-Toledo (Mettler-Toledo S.p. A., Milano, Italy) balance. After the sieving of 1 kg of the material, the different size batches were weighted in order to obtain a mass-based size distribution. 

### 2.5. Size Characterisation by Microscopy and Static Light Scattering

In order to gain more information about the size distribution of the selected PVC sample, the whole sample and the different batches collected with the sieving were analysed by static light scattering using a Malvern Mastersizer 3000 (Malvern Panalytical, Malvern, UK) instrument. 

Particles were suspended in ethanol and added drop by drop to a liquid measurement cell (6 mL total volume) until sufficient obscuration (about 5%) was reached. Each measurement was performed in five replicates and with stirring the ethanolic suspension at 1200 rpm during the analysis. These moderated obscuration and stirring velocity parameters were selected in order to minimise the effect of particle collisions and friction that could lead to particle degradation and reduction in size. The model employed to analyse the raw data considered non-spherical particles and a refractive index set to 1.53 for PVC. 

Optical microscopy analysis of the whole batch was carried out to assess the size distribution using a Zeiss Microscope Axio Imager.M2 microscope (Carl Zeiss S.p.A., Milano, Italy) and applying the ImageJ software with the NanoDefiner tool in order to recognise the particles from the images taken [24,25]. Mass-based size distribution was generated from the number-based data considering spherical particle shape and a particle density of 1.38 g/mL.

### 2.6. Transmission Electron Microscopy (TEM) Analysis

TEM (JEOL JEM-2100, JEOL, Milano, Italy) coupled with EDX (Bruker Italia S.r.L., Milano, Italy) was used at 120 kV in both TEM and STEM modes to characterise the morphology and elemental composition of the PVC-JRCNM70508a particles in the nano-size range. A 3 µL aliquot of the sample suspension was manually deposited on a 200 mesh Formvar (Agar Scientific, Rotherham, UK) carbon-coated copper grid and dried overnight in a desiccator, before analysis.

Elemental analysis was performed in STEM, bright field, and hypermap mode by EDX (Quantax software, Bruker Italia S.r.L., Milano, Italy) to determine the carbon, oxygen, and chlorine content.

### 2.7. Durable Nile Red Staining

Detection of the particles is an essential requisite in this study for the initial and final counting of particles under a fluorescence microscope. For this reason, commercially available fluorescent particles were used and the PVC material from the industrial supplier was stained in-house. This procedure involved a first preparation of the 1 g/L Nile-Red solution in acetone, involving a filtration in order to remove particulates present in it that can be misjudged as particles from the sample [26]. A 100 µL aliquot of this solution was added to 10 mL of a EtOH:H_2_O 1:1 mixture and the solution was mixed with 100 mg of PVC particles in a 30 mL microwave digester (CEM, Cologno Al Serio, Italy) glass vial to further undergo to the staining procedure using an automated microwave digestion system (Microwave Discover SP-D Clinical from CEM, Cologno Al Serio, Italy). The treatment was carried out at 120 °C and 300 psi for 10 min. The final step involved a filtration of the suspension using a 10 µm cut-off Nylon filter (Millipore, Merck, Darmstadt, Germany) and a resuspension of the particles in ethanol.

### 2.8. Spiking

In order to introduce spike particles, MP particles were deposited on a 13 mm KBr pellet and counted using a fluorescent microscope. Subsequently, the KBr pellet was added to the liquid phase. The pellet easily dissolves in any aqueous medium and, in a similar way, particles can be introduced at various steps of the sample treatment procedure including by direct addition to an in-line sampling device by placing the KBr pellet on the (highest cut-off) filter before closing the filtration device. 

### 2.9. Centrifugal Density Separation

After the sampling step, the content of the filters is usually re-suspended in a detergent-containing medium and often treated with acids. This step serves to dissolve iron oxide and calcium and magnesium salt particles, including breaking up possible particle agglomerates. The resulting solution is typically collected in a second filtration step and particles have to be re-suspended in the salt solution needed for density separation. Figure 1 summarises the steps of the density separation procedure. 

In order to model this, a suspension with a total volume of about 20 mL was generated by spiking MilliQ water with MP particles and adding concentrated ZnCl_2_ solution to the suspension. This liquid was transferred in a 35 mL glass microwave digester vessel which was placed in a laboratory centrifuge (Centrifuge 5804 R from Eppendorf S.r.l., Milano, Italy) equipped with a swing rotor. Centrifugation was performed in order to accelerate the flotation of the microplastics and the sedimentation of other particulates (sand, silica, salts, organic materials) having higher density than the medium. Typical values of 1620 rpm for 10 min were applied at 20 °C after previous optimisation using PS particles and Nylon fibres resulting in recovery values in the 82–93% recovery range. 

Subsequently, sedimented contaminants were collected from the bottom part of the glass container, as the MPs remained at the top or adsorbed on the wall of the glass vial. Collection of sedimented particles was performed using a glass pipette connected to a vacuum pump through a vacuum bottle. 

MPs were collected from the remaining (top part) of the suspension, rinsed from the vessel wall, and collected by filtration using a 0.1 µm Anodisc (25 mm) membrane filter. These filters are made of aluminium oxide, thus—in contrast to polymer-based membrane filters—they are not expected to release any microplastic particles. They are transparent for IR light above 1300 cm^−1^ and also have a good transparency in the visible wavelength region, allowing for both FT-IR analysis and visualisation of the particles in transmission mode.

It was proven to be crucial during the full process to rinse several times, with water and ethanol, all glass containers (vials, funnel, pipette) in order to recover the maximum quantity of MPs as they have strong tendency to become attached to the surfaces. Finally, the particles on the filter were counted using the fluorescence microscope and the recovery was calculated according to the equation below (Equation (1)).(1)$$Recovery=\frac{Number\;of\;particles\;collected\;on\;filter\;for\;analysis}{Number\;of\;particled\;deposited\;on\;KBr\;pellet}\times 100\%$$

### 2.10. Pilot Experiments with Fluorescent Polystyrene Particles and Nylon Fibres

Spiking experiments and the recovery study using commercially available fluorescent spherical polystyrene and fibre-shaped Nylon particles were performed without any further treatment of the particles other than suspending them in ethanol. Particles were deposited on a solid support by pipetting an aliquot of the ethanol-based suspension. After drying, they were visualised and quantified sing a fluorescent microscope and dispersed in concentrated ZnCl_2_ before subjecting them to the centrifugal density separation, filtration, and quantification steps.

### 2.11. Density and Viscosity Measurements

Five solutions with different concentrations were prepared starting from a saturated ZnCl_2_ solution applying different dilutions. The density of each ZnCl_2_ solution was determined by mass difference using a glass pycnometer (5 mL nominal volume, Brand, GmbH Wertheim, Germany). The viscosity of the ZnCl_2_ solutions was measured using a Lovis M/ME microviscometer (Anton Paar S.r.l., Rivoli, Italy) working with reduced solution volumes (<1 mL). The instrument provides—based on the Hoeppler principle—the kinetic and dynamic viscosity of a solution of a known density. 

## 3. Results

### 3.1. Characterisation of the PVC Test Material

#### 3.1.1. Morphology and Components

Optical microscopy showed that the particles of PVC-JRCNM70508a have a 30–300 µm size and an irregular shape that often makes it difficult to separate individual particles (Figure 2) even when using the automatic image analysis tool (NanoDefiner) [24]. 

Even if the contribution of small particles to the total particle number seems to be high, the yield in mass is relatively low (Table 1). Still, particles with sizes between 40 and 80 µm can be separated by sieving. This size range is compatible with the suggestions of the Methodology on particles to be applied in recovery tests for testing particles captured on the second (b) 20 µm cut-off filter of the sampling device (Figure 1).

Size fractions generated by sieving from the original batch were also analysed by SLS. As the test material is intended for use in a particle number quantification process, the results from SLS, providing volume-based size distributions, were converted to number-based size distributions (Figure 3). The number-based size distribution of the individual fractions matched well with the size expected based on the cut-off values of the sieves in the lower size range (20–40, 40–80, and 80–160 µm). Meanwhile, results obtained for larger-size fractions (160–315 and 315–600 µm) suggested that the average size of particles is far below the mean of the size limits for the corresponding size classes. The original, unfractionated PVC-JRCNM70508a batch itself exhibits a broad size distribution in SLS analysis, ranging from approximately 60 µm to 300 µm, with the main population concentrated around 100–140 µm. The population of the smallest particles, which were separated by sieving (from 20 to 40 µm), is not detected in the analysis of the original (“Whole”) sample due to the higher concentration and light scattering capacity of the larger particles. 

The TEM analysis of sample PVC-JRCNM70508a showed poly-dispersed particles of irregular morphology, most of them also aggregated/agglomerated with sizes in the micrometric and nanometre ranges (Figure 4). These tiny particles are difficult to separate from the larger ones by the sieving procedure and are difficult to observe by optical microscopy. 

EDX analysis confirmed that the major elemental components of this material were carbon, oxygen, and chlorine (Figure 5). Copper is present in the EDX spectrum due to the application of Formvar carbon-coated copper grids as support and cannot be evaluated as a contaminant based on the EDX results. 

#### 3.1.2. Spectral Fingerprint

The methodology recommended by the EC for the quantification of MPs in water for human consumption foresees the use of spectromicroscopy methods for the identification of particles based on their spectral fingerprint [5]. This includes µRaman, µFT-IR, and also Quantum Cascade Laser spectroscopy (QCL). Thus, the test material selected for method development has to possess clear spectroscopic features that allow for the identification of the particles both with Raman and FT-IR spectroscopy. 

In the preliminary stages of this study, a total of five candidate particulate PVC samples were obtained from industry and analysed by FT-IR and Raman. Most of them showed additional peaks compared to the typical PVC library spectrum. These extra peaks included features belonging to solid additives, for example, titanium-dioxide. The exact formulation of these materials with additives was unknown; some of them appeared to be simple physical mixtures of multiple types of particles. In order to avoid complications related to composition inhomogeneity and non-predictable densities, we selected PVC-JRCNM70508a for further processing that showed the characteristic PVC spectrum in both techniques. The most prominent vibration mode designated to the C–Cl bond is present at 837 cm^−1^ and 637 cm^−1^ in the FT-IR and Raman, respectively. Around 2910 cm^−1^ in the Raman spectrum, the band of C-H stretching appears; and in the range of 1100–1500 cm^−1^, several vibration modes take place (1180 cm^−1^ CH_2_ twist, 1260 cm^−1^ C–H rocking, etc.) The spectra and a table with the principal peaks and their correlations are shown in Figure 6a,b.

Both number-based quantification techniques need sample preparation—the removal of “other” particles from the sample—and the selected preparation method should not interfere with the subsequent spectroscopy-based analysis of the sample. Fluorescent components, for example, might hide the Raman peaks in Raman spectroscopy. Therefore, the selected PVC particles were exposed to the sample preparation procedure presented later and controlled for the presence of their characteristic peaks. The chemical identity of the particles was successfully verified by both spectromicroscopy techniques which also provided a good match with the library reference spectra after the density separation step.

#### 3.1.3. Composition

The chlorine content and the presence of negligible metal contamination of the selected material have been confirmed by various methods. The semi-quantitative analysis by ICP-MS indicated that the chlorine content of the sample was much higher than the value suggested by the local measurement by EDX, but in the right order of magnitude compared to the theoretically expected mass percentage contribution of Cl in pure PVC (i.e., 56.7% *w*/*w*). ICP-MS analysis did not reveal the presence of major metallic additives. The TXRF analysis confirmed these results; no unexpected elements were detected in the sample (Figure 7):

### 3.2. Density Separation Recovery Tests

When designing a density separation step as part of the sample preparation, it is important to bear in mind the complexity and heterogeneity of the possible matrices and MPs that can be found in drinking water. For example, for larger particles, shorter flotation times are needed; on the other hand, higher treatment times might increase the probability of losing the MP particles adsorbed on surfaces or aggregated to other materials. 

Table 2 shows the density and viscosity of various ZnCl_2_ solutions prepared by mixing a ZnCl_2_ solution and water in various ratios. Figure 8 illustrates the correlation between the density and viscosity of the ZnCl_2_ solutions. Considering this, the density separation liquid can be tailored according to the actual needs of the separation process. The parameters used in the density separation experiments with the PVC particles were optimised previously with various analytes, obtaining solid recoveries (82.4 ± 11.1% for 51 µm PS particles and 92.9 ± 3.2% with sand), as well as running theoretical calculations based on the Stokes Law. These results are shown in Appendix A.

We applied final ZnCl_2_ densities of 1.58 g/mL and the centrifugation conditions that are shown in Table 3. Recoveries obtained for the 40–80 µm PVC particle fraction under these conditions were 88.9 ± 4.9 and 89.4 ± 2.7% both in the absence and in the presence of model matrix particles (sand), respectively.

## 4. Discussion

We successfully evaluated a representative PVC material from an industrial batch that showed the compositional characteristics, both in spectroscopy studies and in elemental analysis, expected for the selected polymer. It was also possible to separate a size fraction by dry sieving from the sample which fits the recommendations of the Methodology regarding the test of particle recovery in the 20–100 µm range. Size evaluation of the full sample and the size fractions by various methods showed that it is most probably not possible to avoid large particle agglomeration during dry sieving. As a result, particles smaller than the lower cut-off of the fraction (including nanoparticles) might be still present in each fraction and the mass contribution of the smaller-sized fractions gained by this method is less than expected based on the composition of the sample. On the other hand, it was still possible to separate a good quantity of particles in the size range that could serve as representative test material in density separation studies; and, according to the SLS-based evaluation, less than 10% of the particles (in number) had a size smaller than 31.5 µm.

As a result of the microwave-assisted staining process, our Nile red-labelled PVC particles remained fluorescent in an ethanolic suspension for weeks and were stable in a water-based medium allowing fluorescent microscopy-based observation for the recovery calculations.

We confirmed that applying centrifugation accelerates the flotation of various MP particles and we observed good recovery values both for relatively small-sized and even fibre-shaped microplastics (Appendix A). The use of the glass vial and pipette combined with the previously described procedure permitted the removal of the sediment from the bottom while avoiding contact with plastic tools or components and ensuring the efficient recovery of MP particles from the vial surfaces. The recoveries achieved (>80%) are in line with observations from other density separation studies (performed with much larger, more easily floating particles) [27,28]. Even when combined with another typical sample preparation step of similar recovery, such as acid treatment, these recovery rates would allow the required total recovery to be achieved, as required by the legislation relevant to drinking water analysis in Europe [5].

As all possible parameters of all imaginable MP materials cannot be practically tested, theoretical calculations are essential to determine the relative centrifugal force and time of centrifugation that are needed to obtain a satisfactory separation of the particles. The Stokes equation (Equation (2)) allows for predicting the time (*t*) that particles of a certain diameter (D) and density $\left(\rho_{P}\right)$ need to travel the distance (*l*) from the bottom of a vial to the surface under the applied gravitational force (*g*).(2)$$t=\frac{18\times l\times \eta }{g\times D^{2}\times \left(\rho_{P}-\rho_{M}\right)}$$

As the equation shows, these theoretical calculations need input data not only about the properties of the MP particles possibly present in the sample, but also about the density $\left(\rho_{M}\right)$ and viscosity (η) of the liquid medium. Typical solutions applied in density separation are concentrated salt solutions in the 1.2–1.8 g/mL range such as sodium chloride (1.15–1.32 g/mL), sodium or lithium metatungstate (1.4 and 1.6 g/mL), sodium iodide (1.55–1.86 g/mL), or zinc chloride (1.5–1.87 g/mL), with sodium chloride being the cheapest and most environmentally friendly [29]. However, the density limitation of the NaCl solution does not allow for the flotation-based separation of some common, relatively high-density polymers such as PVC or PET. The use of zinc chloride is more expensive and requires more careful handling, but we selected it as a liquid medium because it provides a colourless solution with a density that allows the separation of these common polymers. It should be noted that the separation of some polymers, such as PTFE (theoretical density 2.2 g/mL), is not possible with any of the above-mentioned salt solutions and PTFE has a density which is higher than some mineral materials such as SiO_2_. On the other hand, higher density comes together with higher viscosity and—as Equation (2) shows—the speed of flotation is inversely proportional to the viscosity of the liquid medium. Thus, at a viscosity five times that of the pure solvent, the flotation time for 20 µm PVC particles on a 10 cm long path, using a liquid medium with a density of 1.58 g/mL, without the application of centrifugal force, would still be more than 3 h. Methods validated with a lower density or larger particles cannot be applied without considering their limitations. For example, the simple zinc chloride solution flotation method presented by Crutsett et al. achieves excellent recoveries for large particles, but would not be applicable for PVC particles smaller than 60 micron with the suggested flotation time [30]. Without calculations using proper density and viscosity input values, density separation equipment users might underestimate the time that proper separation would require under the applied conditions. This leads to undesirably low recovery, especially for smaller particles and materials with higher density.

## 5. Conclusions

We carefully selected and characterised a batch of powdered PVC for application in sample preparation method validation. The separated size fraction of the PVC sample, PVC-JRCNM70508a, after in-house Nile red staining, performed well in our recovery tests that use fluorescent microscopy for particle counting. The Nile red-labelled PVC-JRCNM70508a particles were stable in an ethanol-based suspension, well observable on a solid substrate before adding them to the liquid phase, optically traceable during the separation process when using UV illumination, and well visible under the fluorescent microscope after re-collecting them on the analysis filter. While fluorescent label-based detection does not allow for the chemical identification of microplastic particles and is therefore not recommended for the quantification of microplastics in drinking water, food, or environmental sample analysis, it does allow for much faster evaluation in recovery studies. One of the limitations of the method is that it is not possible to simultaneously evaluate recovery for multiple polymer types when the same labelling dye is used and the particles are similarly irregular in shape. Recovery values close to 90% reached with our irregular-shaped, relatively small and dense particles suggest that even if the step would be combined with other sample preparation elements (acid treatment/oxidative treatment), there is a good probability to have combined recovery levels above the requested 60% [5]. These experiments also show that the presence of other particulates commonly found in drinking water samples, such as sand, should not interfere with the application of the density separation method and the observation of representative test particles. 

Depending on the density of the selected medium, the equipment geometry, and the particle density and size, proper density separation requires various flotation times. The duration of the separation might exceed the time that is convenient for routine analysis if simple gravitational sedimentation (flotation) is used. This is especially true for smaller, higher-density particles. Their recovery might be selectively compromised, if density separation times are not properly set considering their flotation speed under the specific conditions applied. The density and viscosity data of the various ZnCl_2_ solutions presented in Section 3.2 can help laboratories to tailor the density separation procedure to their individual needs. The composition of the salt solution media is presented in Table 2 together with the various mixing ratios of aqueous particle suspension and saturated solution used to produce them. We consider this to be one of the most practical ways to homogenise the particles recovered from a filter in a water-based suspension with the concentrated salt solution to provide a homogenous particle suspension in a high-density liquid. Thus, the table can serve as a guidance for those who need to set up similar protocols.

The actual Methodology describing the quantification of MPs in drinking water provides enough flexibility to allow laboratories to adapt their existing infrastructure and procedures to the common requirements. Our work illustrates how this representative test material can be applied in the testing recovery requirements laid down in the Methodology. With this, we intend to contribute to the more widespread, practical application of the harmonised methodology, which is essential for collecting comparable data, setting regulatory limits, and ensuring that drinking water remains safe for human consumption. In addition, this material can be considered a candidate MP representative material from large-scale industrial production that could be hosted in the JRC Nanomaterials Repository supporting research activities in different fields of application.

## Figures and Tables

**Figure 1 polymers-17-00526-f001:**
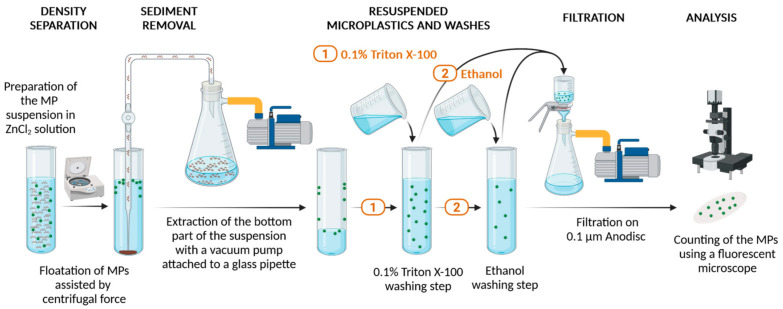
Centrifugal force-assisted density separation of MP particles.

**Figure 2 polymers-17-00526-f002:**
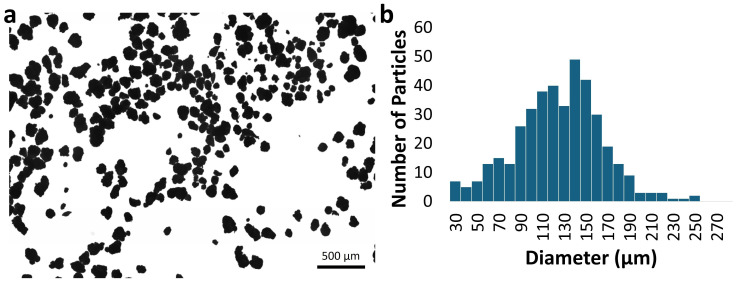
Microscopy image (10× magnification, 3 × 3 stitching) (**a**) and number size distribution generated based on the analysis of PVC-JRCNM70508a particles using the NanoDefiner particle analysis tool (**b**).

**Figure 3 polymers-17-00526-f003:**
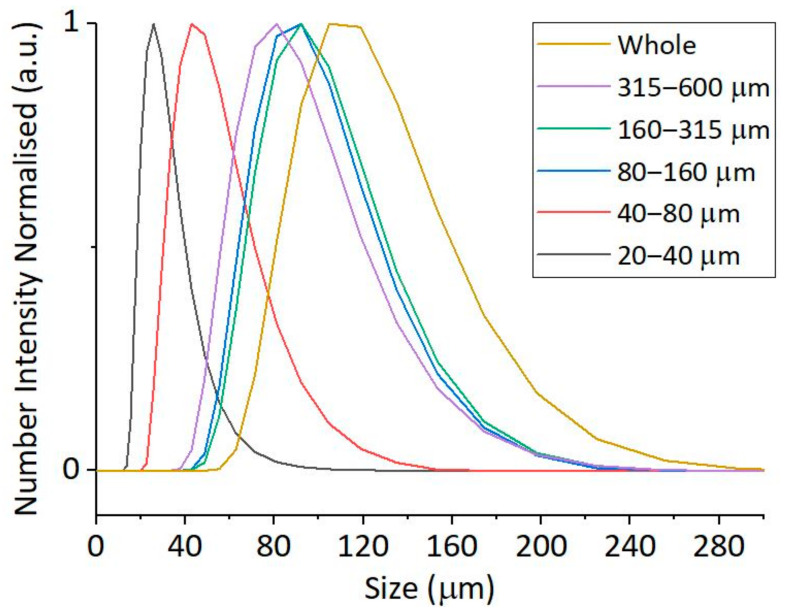
Number-based size distribution of the various PVC-JRCNM70508a fractions generated by sieving. Calculation from SLS measurements. “Whole” labels the unfractionated starting material.

**Figure 4 polymers-17-00526-f004:**
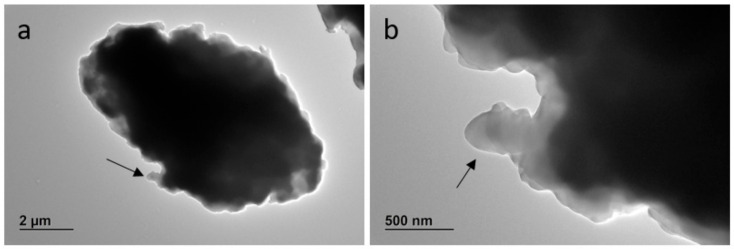
Representative electron microscopy images of PVC-JRCNM70508a particles aggregated/agglomerated on the support grid at 2000X (**a**) and 5000X magnification (**b**). Particles of min Feret less than 500 nm can be also observed (arrow).

**Figure 5 polymers-17-00526-f005:**
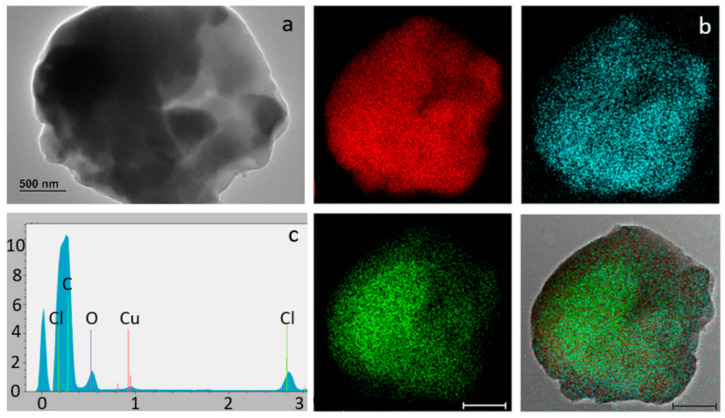
Representative electron microscopy images of PVC-JRCNM70508a (**a**) and the corresponding EDX map, the latter acquired in STEM mode (**b**), scale bars: 700 nm. Carbon (red), oxygen (light blue), and chlorine (green) define the material composition, also confirmed by the cumulative spectra (**c**) of the image; X-axis (KeV), Y-axis (cps/eV); the Cu peak appears due to the support grid.

**Figure 6 polymers-17-00526-f006:**
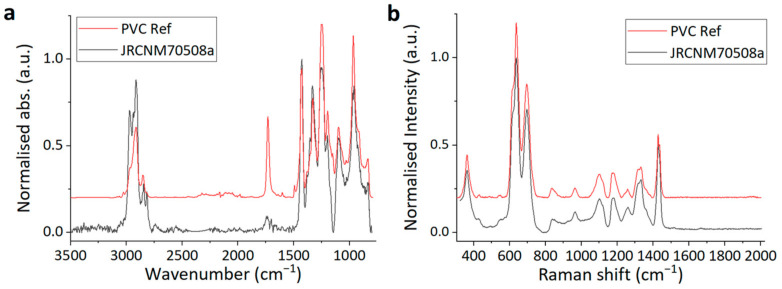
FT-IR (**a**) and Raman (**b**) spectra of PVC sample PVC-JRCNM70508a and a reference spectrum of PVC for each.

**Figure 7 polymers-17-00526-f007:**
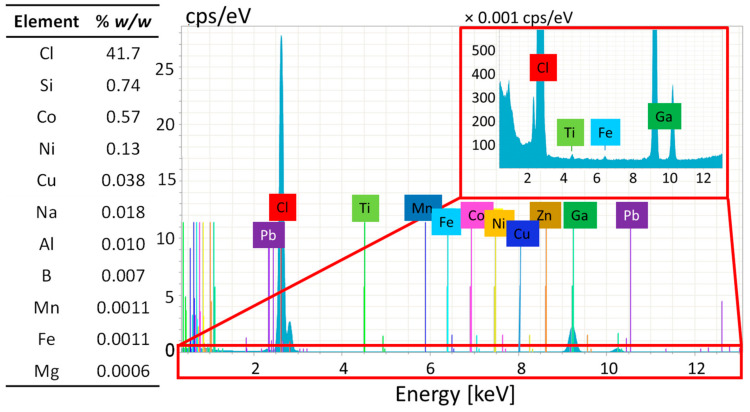
Composition of sample PVC-JRCNM70508a determined by semi-quantitative ICP-MS (**left**) and TXRF spectrum (**right**).

**Figure 8 polymers-17-00526-f008:**
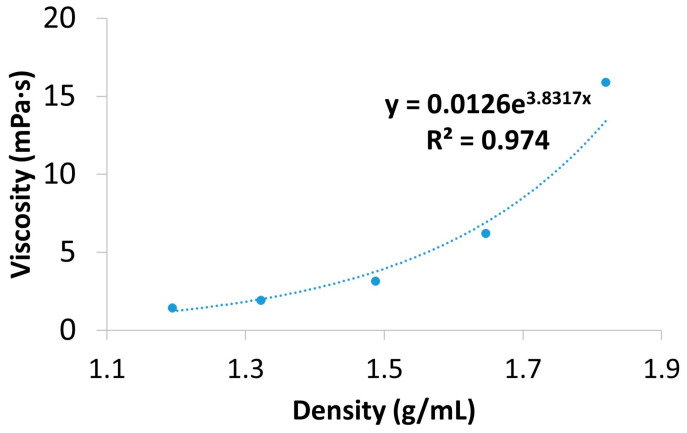
Viscosity and density correlation for ZnCl_2_ solutions at 20 °C.

**Table 1 polymers-17-00526-t001:** Sieve-separated size fractions of sample PVC-JRCNM70508a and their mass-based particle size distribution descriptors D10, D50, D90 determined by SLS.

Size Fraction (µm)	Mass ^a^ %	Mass ^b^ %	D10 (μm)	D50 (μm)	D90 (μm)
315–600	0.1	0.00	57.7	83.0	127.1
160–315	87.3	37.06	67.9	94.1	137.1
80–160	12.3	61.53	65.1	91.0	133.6
40–80	0.3	1.37	31.5	47.0	75.8
20–40	0.0	0.04	19.1	27.5	44.3
Total	100	100.00	82.3	114.5	165.7

^a^ Mass measurement (after Retsch sieving) results. ^b^ Microscopy results based on analysis of “Total” unfractionated sample and after transformation of number to mass distribution.

**Table 2 polymers-17-00526-t002:** Density and viscosity values obtained for different ZnCl_2_ concentrations at 20 °C.

Solution	V_ZnCl_2__ (mL)	V_H_2_O_ (mL)	m (g)	ρ (g/mL)	Viscosity (mPaּ∙s)	Concentration (mol/mL)
1	5	1	9.37	1.82	15.90	0.015
2	4	2	8.48	1.65	6.20	0.012
3	3	3	7.66	1.49	3.15	0.009
4	2	4	6.81	1.32	1.92	0.006
5	1	5	6.15	1.19	1.43	0.003

**Table 3 polymers-17-00526-t003:** Recovery values for various test materials using ZnCl_2_ solutions combined with centrifugal separation.

Analyte	Recovery (%)	SD	Conditions
PVC-JRCNM70508a 40–80 µm	88.9	4.9	1620 rpm 15 min 20 °C
PVC-JRCNM70508a + 10 mg sand	89.4	2.7	1620 rpm 15 min 20 °C

## Data Availability

Raw data available at https://data.jrc.ec.europa.eu/dataset/30df435b-e906-48ab-bdde-91c7f289221f (accessed on 17 January 2025).

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
