# Peer review of "Representative Test Material for Validation of Density Separation as Part of Microplastic Quantification in Drinking Water"

_polymers, 2025, doi:10.3390/polym17040526_

Round 1

Reviewer 1 Report

Comments and Suggestions for Authors

In overall, this paper is presenting information about how the separation and quantification of microplastics are done.

In the materials section, kindly explain the usage of fluorescent polystyrene and nylon fibers.

Kindly explain the observation of Cu in the EDX spectrum.

Kindly confirm whether any possibility of leakage from other of the filter membrane itself of the microplastics and how to avoid it.

Kindly number the equation Recovery.

Kindly explain how to avoid the inter particles breakage and frictional effect cause reduction of size when stirring.

I barely can get the point why density and viscosity of ZnCl2 does matter in this work.

Kindly number which is stoke equation? 

Author Response

We thank to the Reviewers for the comments that identified parts of the text that were not clear for the reader or suggested specific changes that helped to improve the quality of the manuscript. Following the advice from more reviewers, we extended the abstract by adding more specific details on the content, added new points to the discussion and created a Conclusions section and improved the readability of letters in all figures. The changes in the text of the article were marked with red. Please, find below our answers to the individual comments.

Rev1

In overall, this paper is presenting information about how the separation and quantification of microplastics are done.

Q1: In the materials section, kindly explain the usage of fluorescent polystyrene and nylon fibers.

A1: Following the advice of the Reviewer, we inserted a paragraph in the materials section that specifies some additional details of the experiments done with the commercial fluorescent particles. 

Q2: Kindly explain the observation of Cu in the EDX spectrum.

A2: Following the suggestion of the Reviewer, we inserted a sentence in the text explaining that copper is present in the EDX spectrum due to the application of Formvar carbon coated copper grids as support and cannot be evaluated as a contaminant based on the EDX results.

Q3: Kindly confirm whether any possibility of leakage from other of the filter membrane itself of the microplastics and how to avoid it.

A3: The way to avoid the leakage of microplastic from filter membranes is to avoid plastic based membranes during sample preparation and analysis. In fact, the sample treatment protocol suggested in our work describes the use of Anodisc membrane filters. These are made of porous aluminium oxide, thus no leakage of plastic particles from the filter is to be expected. We have to admit, that we applied Nylon filters during the preparation of the in-house stained particles. These filters were not membrane filters, but woven net filters made of very smooth fibres with sharp (thermal) cut at the edges and have the advantage that they are available with larger pore size that helps to avoid the capturing of possible small Nile red dye aggregates together with particles during the staining process. The particle size range targeted in these experiments, the fibre morphology and the quite large pore size of the filters combined with the short duration of the filtration step make it unlikely that this step would produce fluorescent stained Nylon particles in such quantities as to compromise the validity of the recovery experiments carried out with the model PVC particles.

In order to clarify that no leakage of plastic particles is to be expected from Anodisc filters during the sample treatment process, we added a sentence to the manuscript text specifying the material of these membranes and the advantages of their use as follows:

These filters are made of aluminium oxide, thus - in contrast to polymer-based membrane filters - they are not expected to release any microplastic particles. They are transparent for IR light above 1300 cm-1 and have a good transparency also in the visible wavelength region, allowing both FTIR analysis and visualisation of the particles in transmission mode.

Q4: Kindly number the equation Recovery.

A4: We made the equation number better visible in the text by inserting an empty line above the already existing number and inserted a reference to Eq1 in the text.

Q5: Kindly explain how to avoid the inter particles breakage and frictional effect cause reduction of size when stirring.

A5: Stirring has to be applied during the static light scattering analysis, because the density and size of the PVC particles results in fast sedimentation of the particles in the suspending medium. The applied 1200 rpm velocity falls at the medium range in the instrument’s settings and we used quite low particle concentration (obscuration) in order to decrease the effect of particle-particle collisions. We added a sentence to text highlighting that this moderated stirring is recommended in order to avoid the phenomena mentioned by the reviewer.

Q6: I barely can get the point why density and viscosity of ZnCl2 does matter in this work.

A6: Eq2 describes (and it is also mentioned in the text in lines 439-440) that floatation time of particles depends not only on particle size and density but also on the viscosity and density of the surrounding liquid medium. We added now a paragraph to the end of the discussion part that tries to highlight with the help of an example why is it important to properly design density separation experiments and to use these rarely published input parameters in the calculations.

Q7: Kindly number which is stoke equation? 

A7:  We thank to the reviewer for highlighting this. We added the equation number to the Stokes equation and corrected the reference in the text to Eq2.

Reviewer 2 Report

Comments and Suggestions for Authors

This manuscript is more like a experimental report  than a scientific paper. Although the paper contained some experimental results but the discussion of the results was limited. I would like suggest the authors to consider the more research of the actual environmental system and do more work related to this method.

1.The abstract was not background introduction. It need to contain more interesting results of this research.

2. Introduction need to be reorganized. Not more than five paragraphs would be suggested.

3. Why choose  PVC as typical MPs in this study? More types of MPs  should be included. And the results was limited and not convincing.

4. The discussion of the results was too concise. There were not any  interesting finding nor meaningful viewpoint has been put forward.

5. Conclusion was not summarized in this paper. 

Author Response

We thank to the Reviewers for the comments that identified parts of the text that were not clear for the reader or suggested specific changes that helped to improve the quality of the manuscript. Following the advice from more reviewers, we extended the abstract by adding more specific details on the content, added new points to the discussion and created a Conclusions section and improved the readability of letters in all figures. The changes in the text of the article were marked with red. Please, find below our answers to the individual comments.

Rev2

This manuscript is more like a experimental report  than a scientific paper. Although the paper contained some experimental results but the discussion of the results was limited. I would like suggest the authors to consider the more research of the actual environmental system and do more work related to this method.

C1:The abstract was not background introduction. It need to contain more interesting results of this research.

R1:Following the comment of the Reviewer, we added more information in the abstract about the measurement results and experimental content.

C2: Introduction need to be reorganized. Not more than five paragraphs would be suggested.

R2: We reorganized the text and decreased the number of paragraphs to five.

C3: Why choose  PVC as typical MPs in this study? More types of MPs  should be included. And the results was limited and not convincing.

R3: The pilot tests of the centrifugal density separation method were also done by using commercially available, spherical polystyrene and fibre-shaped Nylon particles. This is better highlighted now, as we added a paragraph to the Materials and methods section on this. On the other hand, we kept recovery data of these particles in the Supplementary material to keep focus on the selected PVC material. PVC was chosen for density separation tests because of its relatively high density that is a disadvantage in density separation compared to faster floating, low density materials like polyethylene polypropylene or polystyrene. Following the specific comment of Reviewer 3, additional literature references were added also to the text that explains why PVC was chosen (lines 97-99) also because of its potential presence as secondary microplastic generated from waste in aquatic and terrestrial environments.

C4: The discussion of the results was too concise. There were not any  interesting finding nor meaningful viewpoint has been put forward.

R4: We reorganised the text and added a new, Conclusions section to the article that highlights the achievement of selecting and characterising a representative microplastic test material and the usefulness of the rarely shared input parameters for floatation time estimations.

C5:Conclusion was not summarized in this paper. 

R5: The Conclusion section seemed to be optional in case of “Article” type manuscripts at this Journal.

https://www.mdpi.com/journal/polymers/instructions

However, we received the same advice from another Reviewer as well, thus we reorganised the manuscript text and added a Conclusions section to better highlight the achievement of selecting and characterising a representative microplastic test material, the viability of the microwave assisted fluorescent labelling method and the usefulness of the rarely shared input parameters for floatation time estimations.

Reviewer 3 Report

Comments and Suggestions for Authors

The article described a PVC based representative test material of industrial origin, its physicochem ical characterisation, and application in density separation procedure evaluation, compatibly with the European Commission’s recommendations for quantifying microplastics in water for human consumption. 

I believe this study is meaningful and deserved to be published. 

Author Response

Rev3

The article described a PVC based representative test material of industrial origin, its physicochem ical characterisation, and application in density separation procedure evaluation, compatibly with the European Commission’s recommendations for quantifying microplastics in water for human consumption. 

C1: I believe this study is meaningful and deserved to be published.

R1: We appreciate your review and are grateful for your recommendation to accept our article. Thank you for your time and consideration. 

Reviewer 4 Report

Comments and Suggestions for Authors

Thank you for submitting your manuscript on the evolving regulatory landscape for microplastics, particularly the European Union’s Drinking Water Directive. Your study offers valuable insights into test material selection, characterization, and density separation procedures for microplastics quantification in water.

However, there are several issues that should be improved, including:

1. The abstract is incomplete; it does not clearly present the key findings of the research in terms of both quantity and quality.

2. Some parts of the introduction are unclear, such as the explanation of why the commercially available microplastics considered as "standards" are not suitable.

More references should be added to important content, such as on page 2, line 95, where it states that "Polyvinyl chloride (PVC) was the polymer chosen, due to several key characteristics such as its widespread use worldwide (pipes, packaging, etc.) and consequently its potential presence as secondary microplastic generated from waste in aquatic and terrestrial environments."

Additionally, The factors being studied should be clearly specified.

3. The writing of the model, brand, and country of various testing/analysis instruments is not consistent. Some instruments only have the brand name, while others include the complete information such as model, brand, and country.

4. The figures or text within the figures are all small and unclear, making them difficult to read. Additionally, the font used is inconsistent.

5. In Figure 1, the details of the testing process should be included in the image: preparation of the MP suspension in ZnCl2 solution, flotation of MPs under centrifugation, extraction of the bottom part of the suspension with a vacuum pump attached to a glass pipette, filtration of the remaining supernatant on a 0.1 μm Anodisk filter with washing steps (1 and 2), and counting of the MPs using a fluorescent microscope. The details should not be included in the figure caption to maintain brevity.

6. In Tables 1 and 2, the numbers should be written with the same number of decimal places in the same column.

7.  In Table 3, since both samples used the same centrifuge conditions, it may not be necessary to include this information in the table. Instead, it can be written in the methods section or added to the table title. However, this issue is not of major importance.

8. There is a typo, such as on page 11, line 407, where it says 'Table S3'.

9. There should be a discussion comparing with other methods in previous studies used for microplastic quantification analysis, such as % recovery or others.

10. A "Conclusions" section should be included to summarize the findings of the study.

11. However, I still have concerns about whether this technique can be used for analysis if the sample contains a mixture of different types of plastics. What would the performance be like in such cases?

Additionally, is a % recovery of approximately 88-89% considered relatively low when using this method to analyze substances that may affect public health?

Author Response

Rev4

Thank you for submitting your manuscript on the evolving regulatory landscape for microplastics, particularly the European Union’s Drinking Water Directive. Your study offers valuable insights into test material selection, characterization, and density separation procedures for microplastics quantification in water.

However, there are several issues that should be improved, including:

C1. The abstract is incomplete; it does not clearly present the key findings of the research in terms of both quantity and quality.

R1:The abstract was extended by adding more details about the protocols and results shared in the article.

C2: Some parts of the introduction are unclear, such as the explanation of why the commercially available microplastics considered as "standards" are not suitable.

R2: The sentence about commercially available MPs was revised to explain better the limitations of commercially available particles. Literature references were also added to the text. The actual sentence is this:

Most of the commercially available MPs considered as “standards” are spherical polystyrene (PS) or - less frequently - polyethylene (PE) or poly-methyl methacrylate (PMMA) beads with a density of about 0.8-1.2 g/mL. These few readily available model particles usually do not fulfil the requirements for standard reference materials [10], do not cover the wide range of the physicochemical properties of plastics developed and used in industrial applications (including polymers with densities up to 2.2 g/mL), and do not well represent the irregularly shaped secondary microplastic particles that may be re-leased from these materials.

C 2.2:More references should be added to important content, such as on page 2, line 95, where it states that "Polyvinyl chloride (PVC) was the polymer chosen, due to several key characteristics such as its widespread use worldwide (pipes, packaging, etc.) and consequently its potential presence as secondary microplastic generated from waste in aquatic and terrestrial environments."

R 2.2: Additional references were added to the text, including the paragraph on PVC to strengthen our statement about the industrial use of PVC and its potential presence as secondary microplastic in the environment.

C 2.3:Additionally, The factors being studied should be clearly specified.

R 2.3: The last part of the introduction was modified to specify better the factors studied.

Our work presents not only the strategy we applied during the selection and careful size distribution and composition characterization of the test material, but also the applicability of durable fluorescent labelling for these particles, liquid medium density and viscosity as input parameters and calculations considered when developing and validating our centrifugation-based density separation method, as well as, the results when applying the density centrifugation separation to a variety of different MP types

C3. The writing of the model, brand, and country of various testing/analysis instruments is not consistent. Some instruments only have the brand name, while others include the complete information such as model, brand, and country.

R3: Details on instrument specification (brand, city, country) was added at the necessary points to the materials and methods section of the text.

C4. The figures or text within the figures are all small and unclear, making them difficult to read. Additionally, the font used is inconsistent.

R4: Character type and size in the figures was revised.

C5. In Figure 1, the details of the testing process should be included in the image: preparation of the MP suspension in ZnCl2 solution, flotation of MPs under centrifugation, extraction of the bottom part of the suspension with a vacuum pump attached to a glass pipette, filtration of the remaining supernatant on a 0.1 μm Anodisk filter with washing steps (1 and 2), and counting of the MPs using a fluorescent microscope. The details should not be included in the figure caption to maintain brevity.

R5: We inserted details in the figure and removed from figure caption for brevity.

C6. In Tables 1 and 2, the numbers should be written with the same number of decimal places in the same column.

R6: Appearance of numbers (decimal places) in Table 1 and Table 2 was adjusted.

C7.  In Table 3, since both samples used the same centrifuge conditions, it may not be necessary to include this information in the table. Instead, it can be written in the methods section or added to the table title. However, this issue is not of major importance.

R7: We prefer to keep the conditions column of Table 3 as it is supplemented with Table S3 in Supplementary material, where recovery data for the pilot experiments with PS and Nylon particles are shown with a similar table structure but different parameters.

C8. There is a typo, such as on page 11, line 407, where it says 'Table S3'.

R8: Table S3 is an existing table in the Supplementary material, we checked, both references are correct.

C9. There should be a discussion comparing with other methods in previous studies used for microplastic quantification analysis, such as % recovery or others.

R9: We thank to the Reviewer for the suggestion. We inserted short discussion on this with references in the discussion section.

The recoveries achieved (>80 %) are in line with observations from other density separation studies (performed with much larger, more easily floating particles) [27,28]. Even when combined with another typical sample preparation step of similar recovery, such as acid treatment, they would allow the required total recovery to be achieved, as required by the legislation relevant to drinking water analysis in Europe [5].

C10. A "Conclusions" section should be included to summarize the findings of the study.

R10: The Conclusion section seemed to be optional in case of “Article” type manuscripts at this Journal.

https://www.mdpi.com/journal/polymers/instructions

However, we received the same advice from another Reviewer as well, thus we reorganised the manuscript text and added a Conclusions section to better highlight the achievement of selecting and characterising a representative microplastic test material, the viability of the microwave assisted fluorescent labelling method and the usefulness of the rarely shared input parameters for floatation time estimations.

C11. However, I still have concerns about whether this technique can be used for analysis if the sample contains a mixture of different types of plastics. What would the performance be like in such cases?

R11: We do not suggest the fluorescent labelling method for microplastic analysis, only for the evaluation of recovery from spiked samples during sample preparation procedure validation, as it can be performed faster than spectromicroscopic identification of particles. We agree with the reviewer that it cannot be used for the evaluation of the recovery of multiple polymer types at the same time if the same labelling dye is used and particles are similarly irregular shaped. We added the following sentences to the conclusions to clarify this.

While fluorescent label-based detection does not allow the chemical identification of microplastic particles and is therefore not recommended for the quantification of microplastics in drinking water, food or environmental sample analysis, it does allow much faster evaluation in recovery studies. One of the limitations of the method is that it is not possible to simultaneously evaluate recovery for multiple polymer types when the same labelling dye is used and the particles are similarly irregular in shape.

C12: Additionally, is a % recovery of approximately 88-89% considered relatively low when using this method to analyse substances that may affect public health?

R12: The combined recovery value for the full sample preparation process requested by the EU Legislative Act on the quantification of microplastics in drinking water is ≥60%. We inserted in the conclusion a few sentences with references about comparison with other studies and the value requested in EU legislation.

The recoveries achieved (>80 %) are in line with observations from other density separa-tion studies (performed with much larger, more easily floating particles) [27,28]. Even when combined with another typical sample preparation step of similar recovery, such as acid treatment, they would allow the required total recovery to be achieved, as required by the legislation relevant to drinking water analysis in Europe [5].

Round 2

Reviewer 2 Report

Comments and Suggestions for Authors

I am so sorry about the delay about the review of the revised paper. I have read bout the revised verision of this paper and it has been revised carefully according to the comments. I am glad to suggest the acceptance of this revision of manuscript. Thank you for your kind reminding.

Best wishes,

Qiongjie Wang

26th, Jan, 2025